# Enhanced Anti-Obesity Effects of Euphorbia Kansui Extract through Macrophage and Gut Microbiota Modulation: A Real-World Clinical and In Vivo Study

**DOI:** 10.3390/ph17091131

**Published:** 2024-08-27

**Authors:** Ji-Won Noh, Jung-Hwa Yoo, Byung-Cheol Lee

**Affiliations:** Department of Clinical Korean Medicine, College of Korean Medicine, Graduate School, Kyung Hee University, 26 Kyungheedae-ro, Dongdaemun-gu, Seoul 02447, Republic of Korea; oiwon1002@khu.ac.kr (J.-W.N.); yoojh872@gmail.com (J.-H.Y.)

**Keywords:** anti-obesity, herbal remedies, Euphorbia kansui, insulin resistance, gut microbiota, adipose tissue macrophage, weight management

## Abstract

Rising obesity and associated multi-systemic complications amplify the health burden. Euphorbia kansui (EK) extract is clinically recognized for managing obesity. In a human study, 240 obese individuals were categorized into two cohorts: those receiving solely herbal medicine (HM group) and those administered EK concomitantly with herbal medicine (EK group). An in vivo examination using C57BL/6-*Lep^ob^*/*Lep^ob^* mice elucidated mechanisms involving macrophages and gut microbiota with associated metabolic advantages. The clinical study revealed a significant 7.22% body weight reduction during 91.55 average treatment days and examined 16.71% weight loss at 300 days after treatment. In whole subjects, 60.4%, 21.3%, and 6.3% achieved weight reductions exceeding 5%, 10%, and 15%, respectively. Impressively, the EK group exhibited superior weight loss compared to the HM group (EK: −7.73% vs. HM: −6.27%, *p* = 0.012). The anti-obesity effect was positively associated with EK therapy frequency and herbal medicine duration. In the in vivo study, EK significantly improved insulin sensitivity and mitigated infiltration of adipose tissue macrophages (ATMs) by modulating the CD11c+ and CD206+ subtypes. EK also correlated with increased *Bacteroidetes* and *Firmicutes* populations and reduced *Proteobacteria* and *Verrucomicrobia*. Consequently, EK is an effective adjunctive anti-obesity therapy offering metabolic benefits by modulating ATMs and gut microbiota profiles.

## 1. Introduction

The global attention on obesity is driven by its detrimental effect on a range of health problems; however, there are few effective treatments to target both body weight reduction and improvement of insulin resistance, such as lifestyle, pharmaceutical, and surgical options, which are limited by hard maintenance or adverse effects involving psychological, gastrointestinal, or cardiovascular events [1,2].

The intricate interplay between the accumulation of body fat and its resultant impact on systemic metabolism and immune equilibrium underscores the overarching significance of this issue [3]. Obesity evolves into a chronic inflammatory state with low grade derived by adipose tissue macrophages (ATMs), primarily connecting obesity to the development of insulin resistance, which initiates from aberrant lipolysis and cytokine secretion in adipose tissue [4]. In obese individuals, the precursor monocytes infiltrate adipose tissue in greater numbers, and the ATMs are frequently polarized toward the M1 pro-inflammatory profile, releasing cytokines that impair insulin signaling [5,6]. Recently, the intestinal microbiome has been suggested as a therapeutic target for metabolic problems because the changes in microbial phyla have been reported to affect body composition [7]. High-fat diet feeding changes gut permeability and elevates lipopolysaccharide (LPS) levels, which is essential in triggering infiltration of ATMs. However, weight gain, as well as insulin intolerance, also arise upon modification of intestinal microbiota, irrespective of microbiota-induced LPS production [8]. 

Euphorbia kansui radix (EK), also known as Gan Sui, is a traditional Korean medicinal herb belonging to the Euphorbiaceae family and was first recorded “Shen nong ben cao jing” to attack the water by diarrhea [9]. The therapeutic properties of EK have been reported on cirrhotic ascites [10], pleural effusion [11], diuresis [12], and various cancers [13,14]. The immunomodulatory activities were especially reported, in which EK suppressed Th17 and Th1 differentiation related to excessive inflammation and autoimmune diseases [15]. Van et al. demonstrated that Euphorbia tirucalli, a plant in the Euphorbiaceae family, suppressed CD4+ and CD8+ T cells associated with interleukin-2 and interferon and also inhibited the migration of leukocytes [16]. With respect to metabolic diseases, previous studies revealed that other species in Euphorbiaceae had anti-diabetic [17] and anti-obesity properties [18]; however, there are few studies on EK, though EK therapy has been applied in obese patients in Republic of Korea [19]. Therefore, we collected the clinical data about EK therapy, confirmed the in vivo effects on body weight and insulin resistance, and investigated the mechanisms via monocytes, ATMs, and intestinal microbiota. 

## 2. Results

### 2.1. Clinical Results from the Human Study

#### 2.1.1. Baseline Characteristics

A total of 480 patients who first visited the Weight Management Center to lose weight took herbal medicine (HM) treatment based on the TKM theory. Among them, 240 patients were studied, while the other 240 patients were excluded from the analysis due to overweight, not obesity, or a lack of follow-up visits (Figure 1). 

The number of patients who took EK therapy was 155. Demographic characteristics at baseline did not differ significantly between the EK group and the HM group, including age, height, smoking and drinking habits, and follow-up duration. However, we confirmed the significant tendency of patients with high body weight, BMI, and body fluid to perform EK therapy on the first visit (Table 1). The primary procedure of EK therapy entails inducing diarrhea and vomiting, typically lasting for 6 to 8 h. In the EK group, 58.95% of patients experienced only diarrhea without vomiting, and the number of diarrhea and vomiting was 5.51 ± 2.38 and 0.79 ± 1.24, respectively. The compliance rates of both groups were 68.3 ± 23.5 and 67.2 ± 24.5%, respectively.

#### 2.1.2. Effects on Body Weight

Significant changes in the percentage of body weight reduction were observed from baseline to the final visit of each participant in the trial period (−7.22 ± 5.16%, *p* < 0.001). The impact on weight loss in the EK group was significantly greater than the corresponding effect in the HM group (−7.73 ± 4.98% vs. −6.27 ± 5.40%, *p* = 0.012) (Figure 2a). In total participants, the mean changes in BMI from baseline to each final visit was −2.14 ± 1.85 kg/m^2^ (Figure 2c). The changes in both body weight and BMI demonstrated a statistically significant difference between the EK and HM groups (*p* < 0.001) (Figure 2b,c). In total participants, the average body weight reduction increased over the course of the treatment, and the estimated mean body weight change from baseline to 300 days was −17.6 ± 7.01% (Figure 2d). At 30, 60, and 90 days, the average body weight changes were significantly greater in the EK group compared to the HM group (Figure 2d). The percentages of participants in the EK group who achieved ≥5%, ≥10%, and ≥15% body weight loss were 66.4%, 21.9%, and 7.09%, and those in the HM group were 57.9%, 20.4%, and 6.02%, respectively (Figure 2e). The trial number of EK therapy affected the mean body weight reduction by HM treatment (Figure 2f).

#### 2.1.3. Change in the Body Compositions

In the whole participants, fat weights were significantly changed at the final visit compared to the first visit (−3.35 ± 4.94 kg, *p* < 0.001), but muscle and body fluid were not significantly changed (*p* > 0.05). The changes in fat weight were not significantly different between the two groups (*p* = 0.15); however, the weights of body fluid and muscle decreased in the EK group and increased in the HM group with significant differences (*p* < 0.01) (Figure 3). 

#### 2.1.4. Adverse Events after EK Therapy and during HM Treatment

There was no severe adverse event or unpredictable event after EK therapy, but two patients reported mild abdominal pain until the following day after EK therapy, and one patient reported mild dizziness on the day of EK therapy. There were two patients in the EK group who dropped out because of systolic blood pressure elevation above 200 mmHg and no favorable weight loss, respectively. Any serious adverse event was not reported throughout the duration of HM treatment. Among 200 patients observed at 30 days, 72.5% reported no side effects. The most frequently reported side effect experienced by patients during HM treatment was poor sleep quality; however, its severity was not sufficient to result in treatment discontinuation. Mild constipation, dry mouth, diarrhea, and nausea were the subsequent most commonly reported adverse events. 

### 2.2. Results from Animal Model

#### 2.2.1. Body Weight, Glucose Metabolism, and Lipid Profile

Both the ob/ob and EK groups demonstrated no significant difference in body and epi fat weights. At 4 and 8 weeks, the fasting blood glucose (FBG) of the EK group was significantly lower than that of the ob/ob group (Figure 4). In oral glucose tolerance tests (OGTT), the blood glucose levels at 30, 60, 90, and 120 min were significantly reduced in the EK group compared to the ob/ob group. The area under the curve (AUC) also showed a similar propensity. The EK group showed decreased fasting serum insulin (FSI) and homeostatic model assessment for insulin resistance (HOMA-IR) compared to the ob/ob group (FSI: 2.62 ± 0.58 ng/mL vs. 5.66 ± 1.36 ng/mL, *p* = 0.07; HOMA-IR: 48.30 ± 14.29 vs. 175.13 ± 42.40, *p* < 0.05). Among lipids, triglyceride (TG) and non-esterified fatty acid (NEFA) concentrations in the EK group were significantly decreased compared to the ob/ob group (Figure 4).

#### 2.2.2. Safety Profile

To determine the toxic effect of EK on the liver and kidney function, serum levels of aspartate aminotransferase (AST), alanine aminotransferase (ALT), and creatinine were investigated. The creatinine and AST levels were not influenced by EK administration, and the difference in ALT levels between the EK and ob/ob groups was not significant.

#### 2.2.3. Mechanism of EK Therapy in Macrophages and Monocytes 

A significantly increased percentage of total and CD11c+ ATMs and a decrease in the percentage of CD206+ in the ob/ob group compared to the WT group was demonstrated. The EK group showed a significantly reduced percentage of total and CD11c+ ATMs and an enhanced percentage of CD206+ ATMs (Figure 5). The number of ATMs per epi fat weight was significantly decreased in the ob/ob group compared to the WT group but significantly increased in the EK group compared to the ob/ob group. In monocyte analysis, the ob/ob group had a lower percentage of CD11b^+^/Ly6c^low^ and a higher percentage of CD11b^+^/Ly6c^high^ than the WT group. However, the EK group demonstrated a significantly elevated percentage of CD11b^+^/Ly6c and a decreased percentage of CD11b^+^/Ly6c^high^ compared to the ob/ob group (Figure 6).

#### 2.2.4. Mechanism of EK Therapy in Gut Microbiota

In the phylum analysis, the two major phyla, both *Bacteroidetes* and *Firmicutes*, showed as significantly reduced in the ob/ob group compared to the WT group; however, the EK group demonstrated significant increments in both of them (*Bacteroidetes:* 73.13 ± 0.42% vs. 58.49 ± 2.39%; *Firmicutes:* 18.79 ± 1.64% vs. 13.93 ± 0.55%). *Verrucomicrobia* was the second most abundant in the ob/ob group; however, the proportion of *Verrucomicrobia* substantially decreased in the EK group (2.59 ± 2.16%). The other increasing fecal microbial species in the ob/ob group compared to the WT group were *Actinobacteria* and *Proteobacteria*, both of which were decreased in the EK group. *Deferribacteres* phylum was significantly increased in the EK group compared to the ob/ob group (6.24 ± 1.73% vs. 0.38 ± 0.18%). According to Simpson’s method, the diversity indexes of all three groups were not significantly different from each other. However, the results of Principal Coordinate Analysis (PCoA) and Unweighted Pair Group Method with Arithmetic mean (UPGMA) suggested similar patterns of microbiota composition between the WT and EK groups, distinctly different from the ob/ob group (Figure 7).

## 3. Discussion

The current study was the first real-world clinical study with EK therapy and HM treatment for obesity in Republic of Korea, and it also investigated its mechanism. The patients who took only HM treatment significantly lost 6.27 ± 5.40% of body weight; however, the other patients who additionally took EK therapy once or twice significantly demonstrated additional weight loss of 1.45%. In demographic features, patients in the EK group showed a tendency to have higher BMI, fat, muscle, and body fluid. Moreover, the EK therapy during HM treatment significantly attributed to additional effects on the change in BMI, muscle, and body fluid. According to an in vivo mechanism study, EK therapy showed significant effects on adipose tissue inflammation at macrophage and monocyte levels and changed the gut microbiota towards the pattern of the lean, not that of the obese. Therefore, this article proposed EK therapy as an effective adjunctive treatment to be combined with HM treatment for obesity.

The EK therapy plus HM treatment showed 7.7% body weight and significant reductions in BMI, muscle, and body fluid compared to only HM treatment. In regard to body fat loss, the two groups were not statistically different. As the anti-obesity efficacy increased over the HM treatment time, the EK group showed significantly greater weight loss until 90 days; however, the mean change of body weight at 300 days was greater in the HM group. We confirmed that anti-obesity efficacy significantly increased according to the number of undergoing EK therapies (*p* < 0.01). Notably, the trial number of EK therapy was concentrated during the early treatment period. In the EK group, all subjects underwent EK therapy within 30 days from baseline, and the number of subjects undergoing EK therapy substantially diminished to 13, 10, 6, 5, 3, and 2 at 60, 90, 120, 180, and 300 days, respectively. In a previous prospective study with EK therapy [20], one trial of EK therapy contributed to a significant body weight loss of 1.27 kg. Our HM group was shown as similar to the results from the previous studies treated with various polyherbal decoctions, such as Bofutsushosan [21], Gambisan [22], Taeeumjowee-Tang [23], and Chegamuiyiin-Tang [24]. Gambisan showed 6.2% of body weight loss and 2.87% of body fat. The compliance rate of the Gambisan study was higher than 70%, and the weight loss effect of adjunctive EK therapy was potent, losing 7.73% of body weight and 2.47% of body fat [22]. 

Our study assessed the safety of EK therapy by self-reported adverse events during the treatment period and laboratory blood tests in mice. We observed no toxic effect on the liver and kidney in mice, and Lee et al. [20] confirmed no significant difference in AST, ALT, GGT, BUN, creatinine, and eGFR before and after EK therapy in humans. Also, the adverse events of EK therapy were similar to the results that Lee had reported, such as mild to moderate abdominal pain, which lasted 1–2 days. All our prescriptions for HM treatments included Ma-huang, and no severe adverse effect was reported in common with previous studies, including dry mouth, constipation, diarrhea, palpitation, and insomnia.

The importance of this study is assessing the anti-inflammation effect through various values at the cellular level. In obese patients, persistent, low-grade inflammation is triggered by ATM infiltration and its derived cytokines, which disrupts the insulin signaling cascade-like feedback loop [25]. We investigated the effects of EK on inflammation in this respect. EK therapy decreased the number of ATMs in the quantitative aspect and also significantly improved adipose tissue inflammation in the qualitative aspect. EK treatment significantly decreased CD11C+ ATMs, namely M1 ATMs, which stands for the pro-inflammatory activity of ATMs in adipose tissue, and the same results were shown in the analysis of blood Ly6c monocyte types. Lee also suggested EK-modulated ATMs and pro-inflammatory cytokines, including TNF-α and IL-6, in the same manner as our results.

Ly6c^hi^ monocytes representing pro-inflammatory traits are recruited to the inflammation site by chemo-attractants and differentiate into macrophages or dendritic cells depending on the local cytokines [26]. Ly6c^low^ monocytes are patrolling to differentiate into resident macrophages and promote wound healing [27]. In the obesity inflammation, Ly6c^hi^ and Ly6c^low^ monocytes may differentiate into M1 and M2 macrophages [5,28]. As per our expectation, Ly6c^hi^ in monocyte-related pro-inflammation was decreased, while Ly6c^low^-type of monocyte-related anti-inflammation was increased by EK. 

Alteration of gut microbiota and resultant modulation of intestinal permeability could be a key strategy to ameliorate obesity inflammation [7]. The most definite result in microbiota was the increase in *Verrucomicrobia* in the ob/ob group. *Verrucomicrobia* were counted at 16.85% in the ob/ob group; however, the WT group and EK group showed only 0.00% and 2.59%. The relationship between *Verrucomicrobia* and obesity was reported that high-fat feeding induced the increase in *Verrucomicrobia* and also the increasing ratio of *Verrucomicrobia* to *Bacteroidetes* [29]. Our study showed similar results to this previous study. The mean ratio of *Verrucomicrobia* to *Bacteroidetes* was significantly increased in the ob/ob group; however, the ratio was decreased in the EK group, demonstrating EK had effects on recovering harmful microbial transition. 

In the Genus-level analysis, *Alistipes* belonging to *Bacteroidetes* were significantly decreased in the ob/ob group and recovered in the EK group, even higher than the WT group. *Alistipes* correlate negatively with leukocytes and present more abundantly in healthy subjects compared to non-alcoholic fatty liver disease patients mediated by inflammation [30,31]. Moreover, a human study confirmed that successful weight loss significantly enriched *Alistipes* [31]. Notably, the dominant phyla, *Firmicutes*, was decreased in the ob/ob group and increased in the EK group, contradicting other results. Turnbaugh et al. [32] found an increase in the abundance of *Firmicutes* associated with diet-induced obesity. These compositional changes were completely reversed after returning to a normal diet, which suggests that diet is the main contributing factor to obesity-associated changes in the gut microbiota [33]. In our study, the diets of all experimental groups were identical, and the results of genetically obese and obese individuals with high-fat dietary habits were thought to be different for microorganisms. Indeed, the critical biomarker of obesity is uncertain among *Firmicutes*, *Bacteroidetes*, *Firmicutes*/*Bacteroidetes* ratio, and other phyla and remains to be determined [34]. 

*Deferribacteres* showed a low proportion in both the WT group and the ob/ob group and obviously increased in the EK group. In a study of diet-induced weight modification, *Mucispirillum*, the only genus of *Deferribacteres*, was positively correlated with serum leptin levels [35] and also had several systems for scavenging reactive oxygen species induced by inflammation [36]. Interestingly, *Mucispirillum* was known not to be affected by the prebiotic treatment [37]. Therefore, EK could be a potent option to promote the growth of *Mucispirillum*, finally leading to weight reduction.

We had several limitations. In our pragmatic, real-world clinical study, a disparity in the number of follow-up bioelectrical impedance analyses (BIAs) resulted in limited sufficient analyses in body composition changes, such as fat, muscle, and body fluid, and attributed to the data imbalance between the EK and HM groups. Also, the fidelity of the intervention was restricted due to real-world conditions, lacking placebo or randomization. Second, the compliance of HM treatment and EK therapy was low compared to the other retrospective studies with HM treatment [22]. Additionally, the final body weight was not recorded at the end-of-treatment- but at the last-prescription-date during the study period. These approaches would have underestimated the potential weight loss effects associated with both HM treatment and EK therapy. Third, the numbers of each mouse groups were too small to make a generalization human, and additional studies are needed, with large numbers of animal and human studies. However, our study had enough patients to be the first real-world clinical study, although clinical trials with EK therapy were rare because of its toxicity. In the past, the anti-inflammatory benefits of EK were studied, focusing on various cancers and gastrointestinal tract-associated diseases. However, this study was also the first research of EK in obesity and glucometabolism to investigate potent anti-inflammatory mechanisms of macrophages and Ly6c monocytes, suggesting a positive role in host energy metabolism by the transition of microbiota. Finally, the biomarkers used in this study, such as AST, ALT, and creatinine, are typically only activated when the liver or kidneys have been damaged to some degree; therefore, ultrasensitive PCR methods are needed to detect inflammatory processes that may occur before organ damage. Future studies may also incorporate more sensitive markers than bilirubin, gamma-glutamyl transferase (GGT), liver biopsy, and creatinine to help detect liver and kidney dysfunction at an earlier stage.

In summary, EK therapy, when combined with HM treatment, significantly contributed to additional weight loss and improvements in various body metrics. Our findings from the in vivo mechanism study revealed that EK therapy influenced adipose tissue inflammation at the macrophage and monocyte levels. Specifically, EK therapy reduced the number of pro-inflammatory CD11c+ macrophages (M1 macrophages) and altered the balance between pro-inflammatory Ly6c^hi^ monocytes and anti-inflammatory Ly6c^low^ monocytes, suggesting a modulation of inflammatory responses associated with obesity. Furthermore, we noted significant changes in gut microbiota composition due to EK therapy. EK therapy shifted the microbiota towards a profile more typical of lean individuals, with reductions in the abundance of Verrucomicrobia, which is often elevated in obese conditions, and increased levels of beneficial genera such as Alistipes. These changes in gut microbiota are consistent with previous research linking microbiota modulation to improvements in obesity and metabolic health.

## 4. Materials and Methods

### 4.1. A Real-World Clinical Study

#### 4.1.1. Study Design and Eligible Patients

This study was an open-label, pragmatic cohort study encompassing outpatients who visited the Weight Management Center at Kyunghee University Korean Medicine Hospital (Seoul, Republic of Korea) to treat obesity between 1 January 2022 and 31 December 2022. Eligible patients were included according to the following criteria: (1)Men and women aged 15 years or older with BMI over 25.(2)Those who decided to receive HM treatment among the patients who visited the hospital for weight loss,(3)Each patient who visited the hospital at least twice within the entire treatment period to confirm the change in body weight.

The patients who denied taking HM for weight loss or could not be checked for the follow-up body weight were excluded. All eligible patients underwent HM treatment, of which the contents of the prescription were different for each patient. At the first visit, all patients measured their body weight and performed BIA using InBody720 (InBody Inc., Seoul, Republic of Korea) and were recommended not to take any other medicines or supplements for weight loss during the HM treatment period. 

Patients were fully informed about the potential risks, including diarrhea, vomiting, and abdominal pain, associated with the susceptible EK treatment. We then assigned patients into either the EK treatment group or the HM group based on their choice: the EK group who underwent EK therapy at least once plus conventional HM and the HM group who were prescribed only conventional HM without EK therapy. We compared the changes in body weight and other body compositions before and after HM treatment between the EK group and HM group. For monitoring compliance, we checked how much of the prescribed medication was taken at each patient visit.

This real-world clinical study’s protocol was approved by the Kyung Hee University Korean Medicine Hospital Institutional Review Board (KOMCIRB 2023-07-001), and written informed consent was obtained from all participants.

#### 4.1.2. EK Therapy and HM Treatment

EK was provided by the Department of Pharmaceutical Preparation of the Kyunghee University Korean Medicine Hospital. An EK capsule contains 400 mg of EK powder. EK therapy is performed as a 1-day course at home. Patients were prescribed 8 to 12 EK capsules at a single time, taking 4–5 capsules every 2 min on an empty stomach. After 1–2 h, multiple episodes of diarrhea and vomiting would occur over a period of 6–8 h, along with abdominal pain or nausea. We notified in advance that (1) severe abdominal pain could be relaxed with a hot pack or antispasmodics and (2) dehydration might occur after more than 10 times of diarrhea, and a can of soft drink could be helpful. All participants were prescribed HM, which was *Gami-Samhwang-san* (by decocting Ephedra Herba 4 g, Armeniacae Semen 4 g, Acori Gramineri Rhizoma 4 g, Raphani Semen 4 g, Coicis Semen 4 g, Phellodendri Cortex 4 g, Atractylodes Chinensis Rhizome 4 g, Rhei Radix et Rhizoma 4 g in water), and TKM clinical doctors at the Weight Management Center in Kyunghee University Korean Medicine Hospital modulated the prescriptions according to health problems of each patient. 

#### 4.1.3. Lifestyle Modification

On the first visit, all patients were counseled on correcting their lifestyle based on a common diet and exercise regimen. Our center highlighted taking regular 2 or 3 meals in a day, avoiding fruits and confections, including sweet drinks and sauces, and walking for an hour, 1–2 times a week, until sweating. 

### 4.2. Animal Study

#### 4.2.1. Study Design and Animal and EK Preparations

The experimental animals were wild-type 19~21 g male C57BL/6 mice (WT group, n = 5) and obese 27~31 g male C57BL/6-Lepob/Lepob (ob/ob) mice (Central Lab Animals Inc., Seoul, Republic of Korea). The ob/ob mice were randomly assigned into two groups: the ob/ob group (n = 5) and the EK group (n = 5). Normal chow with 4.5% crude fat and water were supplied ad libitum. The EK group was treated with EK powder (200 mg/kg), orally administered twice a week for 8 weeks, while the WT and ob/ob groups took normal saline. EK was purchased from the Department of Pharmaceutical Preparation of Kyunghee University Korean Medicine Hospital (Seoul, Republic of Korea). All experiments were carried out in accordance with guidelines from the Korean National Institute of Health Animal Facility. The relative humidity in the room was maintained at 40~70%, and the light–dark cycle was set at 12 h. At the end of the study, the mice were euthanized by cervical dislocation.

#### 4.2.2. Weight Measurements and Blood Analysis

Body weights were recorded at the beginning and end of the experiment using the CAS 2.5D scale (Seoul, Republic of Korea) in the morning. The weights of epididymal fat pads were measured at 8 weeks after scarification. To evaluate glucose metabolism, we examined FBG at 2, 4, and 8 weeks using ACCU-CHECK Performa (Australia) and performed OGTT at 6 weeks after glucose (2 g/kg body weight) feeding. Also, FSI was analyzed at 8 weeks using an ELISA reader by 450 nM, and HOMA-IR was calculated using FBG and FSI. We analyzed the lipid profiles, including total cholesterol (TC), high-density lipoprotein cholesterol (HDL-C), low-density lipoprotein cholesterol (LDL-C), TG, and NEFA, as well as the safety profile, including AST, ALT, and creatinine levels, using blood samples collected from the heart at 8 weeks. 

#### 4.2.3. Analysis of ATMs and Monocytes

Stromal vascular cells (SVCs) were segregated from the epi fat samples at the end of the study. The fat samples were soaked in the solution of phosphate-buffered saline (PBS) and 2% bovine serum albumin (BSA) and isolated by collagenase (Sigma, St. Louis, MO, USA) and DNase I (Roche, Brighton, MA, USA). The cell number of SVC was counted by a cellometer (Nexcelom Bioscience LLC., Lawrence, MA, USA). To prepare before fluorescence-activated cell sorting (FACS) analysis, FcBlock (BD Pharmingen, San Jose, CA, USA) was mixed at the ratio of 1:100, and the fluorophore-conjugated antibodies were added to react with the samples as follows: CD45-PerCP-Cy5.5 (Biolgend, San Diego, CA, USA), CD68-APC (Biolgend, USA), CD11c-phycoerythrin (Biolgend, USA), CD206-FITC (Biolgend, USA), CD11b-PE (Biolgend, USA), and Ly6c-APC. All samples were added into FACS tubes after washing with 2% FBS/PBS solution and analyzed by FACS Caliber (BD bioscience, San Jose, CA, USA). We calculated the percentage of macrophages with CD45+/CD68+, CD45+/CD68+/CD206+, CD45+/CD68+/CD11c+, and the percentage of monocytes with CD45+/CD11b+/Ly6c-, CD45+/CD11b+/Ly6c+, and CD45+/CD11b+/Ly6c++ using FlowJo (Tree star Inc., Ashland, OR, USA).

#### 4.2.4. Analysis of the Composition of the Fecal Microbiota

At 8 weeks, the fecal samples were collected, and their DNA was extracted following amplification by polymerase chain reaction (PCR). The Basic Local Alignment Search Tool (*BLAST*) found regions of local similarity between sequences to analyze taxonomic composition for each sample from the Phylum to Species level. In diversity analysis, we used Simpson’s diversity index to calculate alpha diversity. PCoA and UPGMA clustering trees were used to assess the variation.

### 4.3. Statistical Analysis

The continuous variables were reported in the clinical study using the mean and standard deviation (SD), presented as mean ± SD, and in the animal study using the mean and standard error of the mean (SEM), presented as mean ± SEM. We compared the continuous variables between the EK group and the other group using the Mann–Whitney U test only for baseline age and height using the student’s *t*-test. Linear regression was performed to analyze the changes in weight loss between the EK and HM treatments using linear regression. In our model, we included age, gender, and treatment duration as covariates to control for these potential confounding factors.

We conducted an analysis of variance (ANOVA) and Tukey’s post hoc to compare the weight loss outcomes across the number of EK therapy sessions in the clinical study and to analyze group differences in the animal study. The change in body weight before and after obesity treatment was analyzed by the Wilcoxon signed rank test. We also used a student’s *t*-test to check if a final weight loss was different between different subgroups of gender and behavioral factors. All statistical analyses were performed using PRISM 7 (Graphpad software Inc., La Jolla, CA, USA). All *p* values were two-tailed, and significance was set at *p* < 0.05.

## 5. Conclusions

We can infer the effect of Euphorbia kansui treatment on the grounds of the results related to anti-inflammation and alteration of microbial community composition. These different types of microorganisms might be linked to changes in nutrient absorption and energy regulation. Additionally, we can also assert that leading changes in adipose tissue macrophage and blood Ly6c^hi^ monocytes level by Euphorbia kansui treatment can reduce systemic inflammation activity and finally impact the insulin signaling network. 

## Figures and Tables

**Figure 1 pharmaceuticals-17-01131-f001:**
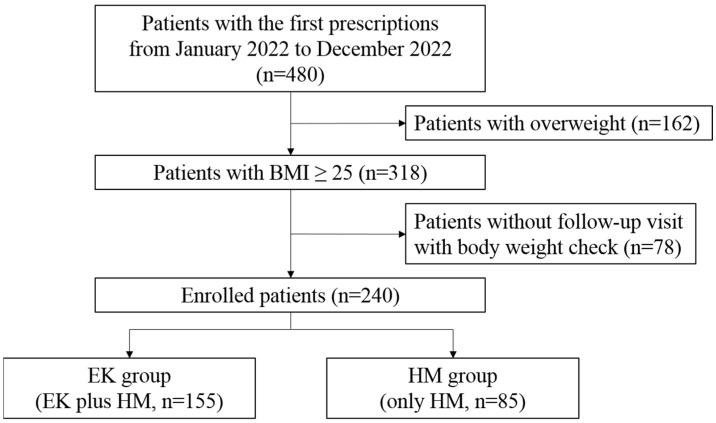
The flow chart of clinical study. BMI, body mass index; EK, Euphorbia kansui; HM, herbal medicine.

**Figure 2 pharmaceuticals-17-01131-f002:**
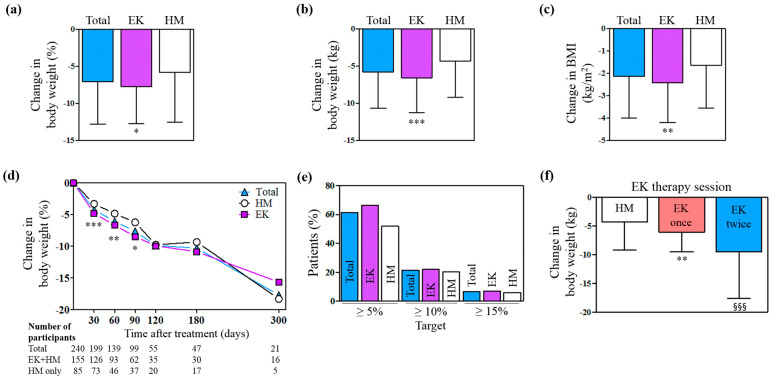
Anti-obesity outcomes of EK therapy and HM treatment. Change in body weight by (**a**) percentage and (**b**) kg, and (**c**) BMI from baseline to the final visit. (**d**) Mean changes from baseline in body weight. (**e**) Proportions of participants achieving body weight reductions of at least 5, 10, and 15%. (**f**) Mean body weight changes according to the trial number of EK therapy throughout the treatment period. * *p* < 0.05; ** *p* < 0.01; *** *p* < 0.001 versus HM group. ^§§§^
*p* < 0.001 versus the EK once group. EK, Euphorbia kansui; HM, herbal medicine; BMI, body mass index.

**Figure 3 pharmaceuticals-17-01131-f003:**
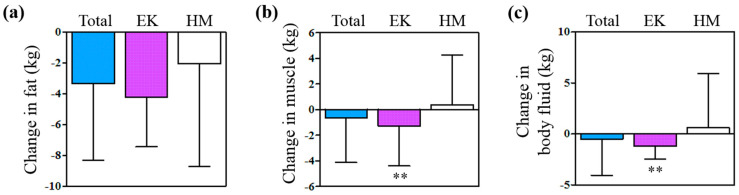
Change in (**a**) fat weight, (**b**) muscle weight, and (**c**) body fluid. Subgroup analysis divided by the number of EK therapy trials. ** *p* < 0.01 versus the HM group. EK, Euphorbia kansui; HM, herbal medicine.

**Figure 4 pharmaceuticals-17-01131-f004:**
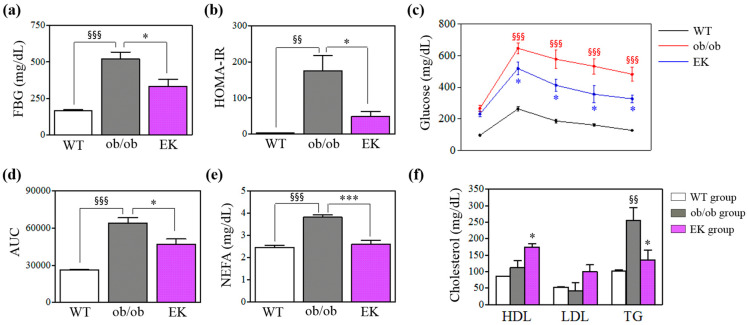
Metabolic benefits of Euphorbia kansui in the animal study. Comparison between three groups in (**a**) FBG, (**b**) HOMA-IR, (**c**) oral glucose tolerance test (OGTT) results, (**d**) AUC of OGTT, (**e**) NEFA, and (**f**) cholesterols. * *p* < 0.05; *** *p* < 0.01 versus the ob/ob group. ^§§^
*p* < 0.01; ^§§§^
*p* < 0.001 versus the WT group. FBG, fasting blood glucose; WT, wild-type; EK, Euphorbia kansui; HOMA-IR, homeostatic model assessment for insulin resistance; AUC, area under the curve; NEFA, non-esterified fatty acid; HDL, high-density lipoprotein; LDL, low-density lipoprotein; TG, triglyceride.

**Figure 5 pharmaceuticals-17-01131-f005:**
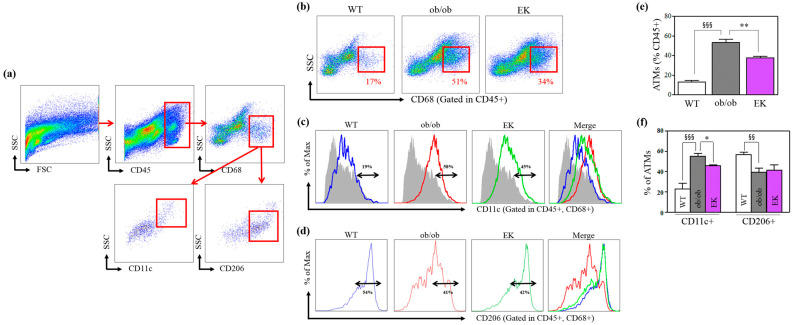
Anti-inflammatory changes in adipose tissue macrophages. (**a**) Flow cytometry results showing adipose tissue macrophage gating (CD45+ leukocytes, CD68 + macrophages, CD11C + macrophage and CD206 macrophages). (**b**–**f**) Percentages of adipose tissue macrophages in total, CD11c+, and CD206+ subtypes of WT (blue), ob/ob (red) and EK (green line). Gray shadow indicates fluorescence minus one control. * *p* < 0.05; ** *p* < 0.01 versus the ob/ob group. ^§§^ *p* < 0.01; ^§§§^ *p* < 0.001 versus WT group. WT, wild-type; EK, Euphorbia kansui; ATMs, adipose tissue macrophages.

**Figure 6 pharmaceuticals-17-01131-f006:**
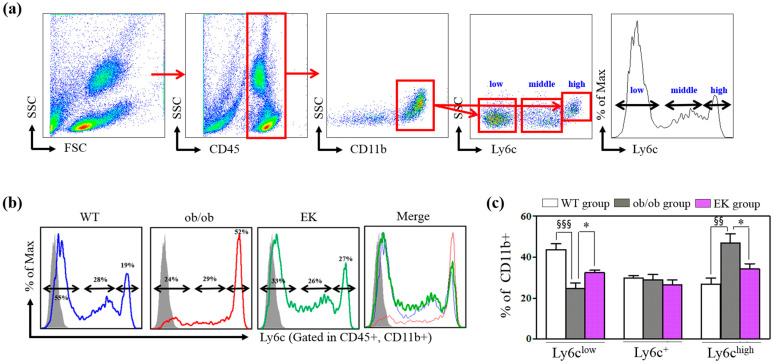
Anti-inflammatory changes in monocytes. (**a**) Flow cytometry showing monocyte gating (CD45+ leukocytes, CD11b + monocytes, and Ly6C low, middle and high subtypes monocytes). (**b**,**c**) Populations of Ly6^−^, Ly6c^+^, and Ly6c^++^ monocytes of WT (blue), ob/ob (red) and EK (green line). Gray shadow indicates fluorescence minus one control. * *p* < 0.05 versus the ob/ob group. ^§§^ *p* < 0.01; ^§§§^ *p* < 0.001 versus WT group. WT, wild-type; EK, Euphorbia kansui.

**Figure 7 pharmaceuticals-17-01131-f007:**
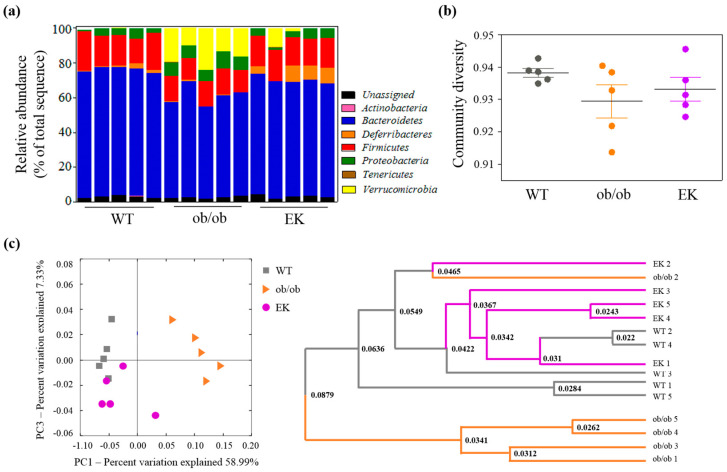
The changes in gut microbiota structure and diversity. (**a**) Percentage of each phylum in the gut microbiota. (**b**) Diversity of gut microbiota. (**c**) Principal coordinate analysis (PCoA) analysis of individuals and UPGMA clustering. WT, wild-type; EK, Euphorbia kansui.

**Table 1 pharmaceuticals-17-01131-t001:** Baseline characteristics of enrolled patients.

Variables	Total	EK Group ^1^	HM Group	*p* Value
N	240	155	85	
Age (years)	41.28 ± 12.17	41.75 ± 12.20	40.42 ± 12.14	0.442
Male:Female (%)	29.58:70.42	34.19:65.80	21.18:78.82	0.034
Height (cm)	164.6 ± 8.41	165.4 ± 8.78	163.3 ± 7.52	0.062
Body weight (kg)	81.25 ± 14.73	84.75 ± 15.38	74.88 ± 10.97	<0.001
BMI (kg/m^2^)	29.84 ± 4.12	30.85 ± 4.34	28.01 ± 2.94	<0.001
Follow-up duration (days)	91.55 ± 66.77	93.30 ± 70.43	88.36 ± 59.78	0.903
Fat weight (kg)	32.27 ± 15.75	33.93 ± 17.89	28.31 ± 7.49	<0.001
Muscle weight (kg)	27.91 ± 6.41	28.81 ± 6.60	25.80 ± 5.42	<0.001
Body fluid (kg)	36.95 ± 7.76	38.09 ± 7.96	34.26 ± 6.58	<0.001
Smokers (%)	16.35	18.71	12.05	
Non-drinkers (%)	41.66	38.06	51.81	

^1^ Patients in the EK group received EK and HM treatments. *p* values show the results from Mann–Whitney U test and student’s *t*-test between EK and HM groups. BMI, body mass index.

## Data Availability

Data is contained within the article.

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
