# Peer review of "Enhanced Anti-Obesity Effects of Euphorbia Kansui Extract through Macrophage and Gut Microbiota Modulation: A Real-World Clinical and In Vivo Study"

_pharmaceuticals, 2024, doi:10.3390/ph17091131_

Round 1

Reviewer 1 Report

Comments and Suggestions for Authors

Enhanced Anti-Obesity Effects of Euphorbia Kansui Extract hrough Macrophage and Gut Microbiota Modulation: A Real-World Clinical and In Vivo Study.

Strengths of the study:

1. It is a study that uses the extract of Euphorbia kansui (EK) that has been used for the management of obesity, on the one hand, in obese individuals, as well as in mice of the C57BL/6-Lepob/Lepob strain that was useful to elucidate the mechanisms that involve macrophages and the intestinal microbiota.

2.      The clinical study revealed a significant reduction in body weight 300 days after treatment, moreover, EK in vivo improved insulin sensitivity and mitigated macrophage infiltration of adipose tissue.

3.    EK was also correlated with increased populations of Bacteroidetes and Firmicutes and reduced Proteobacteria and Verrucomicrobia, suggesting that EK is an effective complementary anti-obesity therapy that offers metabolic benefits by modulating ATM23 as well as gut microbiota profiles.

The areas of opportunity (weaknesses) that need to be addressed by the authors to strengthen the manuscript are mentioned below point by point:

1.    Why does Table 1 of the results include only the percentage of men, and not that of women, since the design mentioned that women were also included?.

2.    Why was the ultrasensitive PCR test, which is very useful in inflammatory processes, not included in the Safety profile?.

3.    If it is known that creatinine is a poor renal marker, since this parameter is elevated when there is already declared renal damage, Cystatin C or CKD- EPI or microalbuminuria would have been used instead.

4.    BUN and eGFR were not mentioned in the results shown (safety profile). Clarification.

5.    In Table 1 of results only the percentage of men is mentioned, what happened to the women participating in this study?.

6.    Section 4.1.2.EK therapy and HM treatment mentions the following: dehydration might occur after more than 10 times of diarrhea and a can of soft drink could be helpful, This seems incoherent to me, since in the first place the selected patients already had obesity problems and surely with alterations in glucose and lipid values, and in addition to this, to alleviate the side effects of the herbal extract they would have to drink a can of soda, which is to a certain extent counterproductive. To clarify.

7.    In section 4.1.3.Lifestyle modification the following is mentioned: sweet drinks, Contradictory to what is mentioned in the paragraph above.To clarify.

8.    In section 4.2.1. Study design, animal and EK preparation the following information is required: Age of mice, weight, what diet they were given to gain weight, shelter conditions including temperature and light/dark cycles, describe the route of administration and what it was administered with, care during the experiment, were they euthanized? If so, how?.

9.    In section 4.2.2. Weight measurements and blood analysis, the following information is required: Were they measured for waist, snout-anus and snout-tail length? If so, what was this done with?. What anesthetic was used (complete details including pharmaceutical laboratory) and in what dose?.

Author Response

Strengths of the study:

  1. It is a study that uses the extract of Euphorbia kansui (EK) that has been used for the management of obesity, on the one hand, in obese individuals, as well as in mice of the C57BL/6-Lepob/Lepob strain that was useful to elucidate the mechanisms that involve macrophages and the intestinal microbiota.
  2. The clinical study revealed a significant reduction in body weight 300 days after treatment, moreover, EK in vivo improved insulin sensitivity and mitigated macrophage infiltration of adipose tissue.
  3. EK was also correlated with increased populations of Bacteroidetes and Firmicutes and reduced Proteobacteria and Verrucomicrobia, suggesting that EK is an effective complementary anti-obesity therapy that offers metabolic benefits by modulating ATM23 as well as gut microbiota profiles.

The areas of opportunity (weaknesses) that need to be addressed by the authors to strengthen the manuscript are mentioned below point by point:

Comment 1. Why does Table 1 of the results include only the percentage of men, and not that of women, since the design mentioned that women were also included?.

A) As you commented, we make an additional line addressing the percentage of women.

Comment 2. Why was the ultrasensitive PCR test, which is very useful in inflammatory processes, not included in the Safety profile?

A) As you commented, ultrasensitive PCR tests play a crucial role in various safety-related fields by detecting extremely low levels of genetic material and are a very useful method for evaluating inflammation in safety profile. However, in this study, we analyzed serum levels of aspartate aminotransferase (AST), alanine aminotransferase (ALT), and creatinine for the safety profile because these biomarkers are well-established indicators of liver and kidney function. Monitoring AST and ALT levels helps assess potential liver toxicity, while creatinine levels are used to evaluate kidney function. These measures provide a comprehensive understanding of the safety profile in the context of potential organ damage, which was the primary focus of our safety assessment.

Comment 3. If it is known that creatinine is a poor renal marker, since this parameter is elevated when there is already declared renal damage, Cystatin C or CKD- EPI or microalbuminuria would have been used instead.

A) Although creatinine might be a poor parameter with pre-existing renal dysfunction, we convinced that creatinine would be a suitable marker because the subjects used in this study were wild-type male C57BL/6 mice entering the intervention period with EK powder just after 1 week for adaptation.

Comment  4. BUN and eGFR were not mentioned in the results shown (safety profile). Clarification.

A) We didn’t analyzed BUN and eGFR, however, we only referred the other human study which confirmed no significant chance in BUN and eGFR after EK therapy.

- Line 213: Lee et al. [18] confirmed no significant difference in AST, ALT, GGT, BUN, creatinine and eGFR before and after EK therapy in human.

Comment 5. In Table 1 of results only the percentage of men is mentioned, what happened to the women participating in this study?.

A) As you commented, we additionally mention the percentage of women in Table 1.

Comment 6. Section 4.1.2.EK therapy and HM treatment mentions the following: dehydration might occur after more than 10 times of diarrhea and a can of soft drink could be helpful, This seems incoherent to me, since in the first place the selected patients already had obesity problems and surely with alterations in glucose and lipid values, and in addition to this, to alleviate the side effects of the herbal extract they would have to drink a can of soda, which is to a certain extent counterproductive. To clarify.

A) EK therapy intentionally induces the response of several diarrhea and mild vomiting. The dose of EK in this study is usually at low risk of dehydration, but the degree of reaction may vary from individual to individual.

Comment 7. The recommendation to drink a can of soft drink (less than 240ml) is the first aid necessary for situations where dehydration is suspected due to excessive diarrhea more than 10 times.

In section 4.1.3.Lifestyle modification the following is mentioned: sweet drinks, Contradictory to what is mentioned in the paragraph above.To clarify.

A) First of all, EK therapy is 1 day course and as we answered above, the recommendation to drink a can of soft drink (less than 240ml) is the first aid necessary for situations where dehydration is suspected due to excessive diarrhea more than 10 times.

Comment 8. ‘4.1.3. Lifestyle modification’ is the content of commonly controlled life therapy education for all groups.

In section 4.2.1. Study design, animal and EK preparation the following information is required: Age of mice, weight, what diet they were given to gain weight, shelter conditions including temperature and light/dark cycles, describe the route of administration and what it was administered with, care during the experiment, were they euthanized? If so, how?.

A) The used animals were 6-week-old male mice weighing 19~21g of wild-type and 27~31g of obese-type. A normal chow with 4.5% crude fat, and water were supplied ad libitum. The relative humidity in the room was maintained at 40~70% and the light-dark cycle was set at 12 hours. The mice were euthanized by cervical dislocation.

Comment 9. In section 4.2.2. Weight measurements and blood analysis, the following information is required: Were they measured for waist, snout-anus and snout-tail length? If so, what was this done with?. What anesthetic was used (complete details including pharmaceutical laboratory) and in what dose?.

A) We didn’t measure the lengths of waist, snout to anus and snout to tail.

Reviewer 2 Report

Comments and Suggestions for Authors

This article, titled " Enhanced Anti-Obesity Effects of Euphorbia Kansui Extract 2 through Macrophage and Gut Microbiota Modulation: A Real-3 World Clinical and In Vivo Study," offers valuable contributions to the field of obesity research. It provides clinical and mechanistic insights into the effects of Euphorbia kansui extract. In summary, the results are encouraging and justify future exploration of the possibility of EK extract as a viable treatment for obesity.

Minor Revision

1.     The introduction lacks review of Euphorbia kansui radix (EK) mainly phytochemical and nutritional composition. The study clearly lacking supporting data behind the anti-obesity effects

2.     Please check as I don’t see any ethical approval if necessary regarding patient consent and the potential risks associated with the EK therapy.

3.     How the compliance was monitored is a real question behind this sort of study

4.     More information on randomization and confounder information would improve the study's validity.

5.     The reported 58% incidence of diarrhea is quite high. This needs to be addressed thoroughly in the manuscript, both in terms of its implications for patient adherence and overall treatment viability.

6.     Please mention patients visited twice per week or per month?

7.     The use of appropriate statistical methods to compare weight loss between groups is appreciated. Including a more comprehensive statistical analysis (e.g., regression models) to control for potential confounding variables would be beneficial.

8.     The discussion section is lacking the mechanism by which EK therapy might exert its anti-obesity effects, particularly through macrophage and gut microbiota modulation.

The study highlights the anti-obesity properties of Euphorbia Kansui extract through macrophage and gut microbiota modulation, and with revisions, it will be well-positioned for publication.

Comments on the Quality of English Language

no comments

Author Response

This article, titled " Enhanced Anti-Obesity Effects of Euphorbia Kansui Extract 2 through Macrophage and Gut Microbiota Modulation: A Real-3 World Clinical and In Vivo Study," offers valuable contributions to the field of obesity research. It provides clinical and mechanistic insights into the effects of Euphorbia kansui extract. In summary, the results are encouraging and justify future exploration of the possibility of EK extract as a viable treatment for obesity.

Minor Revision

Comment 1. The introduction lacks review of Euphorbia kansui radix (EK) mainly phytochemical and nutritional composition. The study clearly lacking supporting data behind the anti-obesity effects

A) It is the fact that there is lack of data for the anti-obesity effects of EK, however, as we described in line 58, in clinic, EK had been prescribed to patient with abnormal weight gain.

Comment 2. Please check as I don’t see any ethical approval if necessary regarding patient consent and the potential risks associated with the EK therapy.

A) In ‘4.1.1.’ section, “This real-world clinical study was approved its protocol by Kyung Hee University Korean Medicine Hospital Institutional Review Board (KOMCIRB 2023-07-001) and the written informed consent was obtained from all participants.”

Comment 3. How the compliance was monitored is a real question behind this sort of study

A) As your comments, in real-world clinical studies, monitoring compliance is essential to ensure that participants adhere to the prescribed treatment protocol. In this study, we checked how much of the prescribed medication was taken at each patient visit for monitoring compliance. In ‘4.1.1.’ section, we add the description as follows; For monitoring compliance, we checked how much of the prescribed medication was taken at each patient visit.

Comment 4. More information on randomization and confounder information would improve the study's validity.

A) In this real world clinical study, participants who consented to the study were informed of the diarrhea, vomiting, and abdominal pain that may occur in the susceptible treatment arm and were assigned to the EK treatment group or control HM group based on patient choice.

In this real-world clinical study, we opted for a non-randomized design to better reflect clinical practice, where participants have the autonomy to choose their treatment group. Participants were fully informed about the potential risks, including diarrhea, vomiting, and abdominal pain, associated with the susceptible EK treatment. They were then assigned to either the EK treatment group or the control HM group based on their choice. To mitigate potential biases, we conducted an assessor-blinded analysis of objective endpoints and additional animal experiment to control for confounding factors. In ‘4.1.1.’ section, we add the description as follows; Patients were fully informed about the potential risks, including diarrhea, vomiting, and abdominal pain, associated with the susceptible EK treatment. We then assigned patients into either the EK treatment group or the HM group based on their choice

Comment 5. The reported 58% incidence of diarrhea is quite high. This needs to be addressed thoroughly in the manuscript, both in terms of its implications for patient adherence and overall treatment viability.

A) EK is a traditional herb which attack the water, so diarrhea could be one of the reactons after having EK. We added some information of EK in line 49, “Gan Sui is first recorded “Shen nong ben cao jing” to attack the water (逐水) by diarrhea {Hou, 2014 #74}”.

Please mention patients visited twice per week or per month?

A) Each patient who visited the hospital at least twice within the entire treatment period to confirm the change of body weight.

Comment 6. The use of appropriate statistical methods to compare weight loss between groups is appreciated. Including a more comprehensive statistical analysis (e.g., regression models) to control for potential confounding variables would be beneficial.

A) we added the statistical methods; We analyzed the changes in weight loss between the EK and HM treatments using linear regression. In our model, we included age, gender, and treatment duration as covariates to control for these potential confounding factors.

Comment 7. The discussion section is lacking the mechanism by which EK therapy might exert its anti-obesity effects, particularly through macrophage and gut microbiota modulation.

A) we added the mechanism of EK therapy in discussion section; In summary, EK therapy, when combined with HM treatment, significantly contributed to additional weight loss and improvements in various body metrics. Our findings from the in vivo mechanism study revealed that EK therapy influenced adipose tissue inflammation at the macrophage and monocyte levels. Specifically, EK therapy reduced the number of pro-inflammatory CD11c+ macrophages (M1 macrophages) and altered the balance between pro-inflammatory Ly6chi monocytes and anti-inflammatory Ly6clow monocytes, suggesting a modulation of inflammatory responses associated with obesity. Furthermore, we noted significant changes in gut microbiota composition due to EK therapy. EK therapy shifted the microbiota towards a profile more typical of lean individuals, with reductions in the abundance of Verrucomicrobia, which is often elevated in obese conditions, and increased levels of beneficial genera such as Alistipes. These changes in gut microbiota are consistent with previous research linking microbiota modulation to improvements in obesity and metabolic health.

The study highlights the anti-obesity properties of Euphorbia Kansui extract through macrophage and gut microbiota modulation, and with revisions, it will be well-positioned for publication.

Reviewer 3 Report

Comments and Suggestions for Authors

The manuscript “Enhanced anti-obesity effects of Euphorbia Kansui extract through macrophage and gut microbiota modulation: A real-world clinical and in vivo study” is devoted to investigation of anti-obesity effects of traditional Korean medicinal herb extract. The authors collected the clinical data about Euphorbia Kansui extract therapy, confirmed the in vivo effects on body weight and insulin resistance, and investigated the mechanisms via monocytes, adipose tissue macrophages and intestinal microbiota.

The manuscript is well written and sufficiently illustrated. The presented data relate to a current problem of our time – obesity. My comments for the authors:

- Line 57. The authors claimed that “there are few studies on EK though EK therapy has been applied in obese patients in South Korea.” Please provide references to the studies you mention.

- The captions to many of the figures do not provide explanations of the abbreviations.

- The authors should pay more attention to the description of Euphorbia Katsui extract. The authors only mentioned that “EK was provided from the Department of Pharmaceutical preparation of Kyunghee University Korean Medicine Hospital. EK capsule contains 400 mg of EK powder.” It is unclear what extract the authors used. Is this a registered drug? Then the name should be indicated. From what part of Euphorbia Kansui was this extract obtained? Does the capsule contain 400 mg of Euphorbia Kansui extract only or together with fillers?

- The same applies to the drug Gami-Samhwang-san. Please provide the plant species name in Latin for a complete understanding of the composition of the preparation. What is the content of active ingredients in this drug?

- It is not clear from the manuscript why the authors chose Gami-Samhwang-san as a comparison drug. Perhaps it would be worth adding in the Introduction references for the use of this drug or its components in the treatment of obesity.

- What dosage of the drug Gami-Samhwang-san was used?

Author Response

The manuscript “Enhanced anti-obesity effects of Euphorbia Kansui extract through macrophage and gut microbiota modulation: A real-world clinical and in vivo study” is devoted to investigation of anti-obesity effects of traditional Korean medicinal herb extract. The authors collected the clinical data about Euphorbia Kansui extract therapy, confirmed the in vivo effects on body weight and insulin resistance, and investigated the mechanisms via monocytes, adipose tissue macrophages and intestinal microbiota.

The manuscript is well written and sufficiently illustrated. The presented data relate to a current problem of our time – obesity. My comments for the authors:

Comment 1.  Line 57. The authors claimed that “there are few studies on EK though EK therapy has been applied in obese patients in South Korea.” Please provide references to the studies you mention.

A) As your comment, we additionally referred Kim’s study which described the safety of EK therapy in patients with abnormal weigh gain.

Comment 2.  The captions to many of the figures do not provide explanations of the abbreviations.

A) Thank you for your comment. We added abbreviations in all captions.

Comment 3.  The authors should pay more attention to the description of Euphorbia Katsui extract. The authors only mentioned that “EK was provided from the Department of Pharmaceutical preparation of Kyunghee University Korean Medicine Hospital. EK capsule contains 400 mg of EK powder.” It is unclear what extract the authors used. Is this a registered drug? Then the name should be indicated. From what part of Euphorbia Kansui was this extract obtained? Does the capsule contain 400 mg of Euphorbia Kansui extract only or together with fillers?

A) EK is an herb that is classified as an herbal medicine by the Ministry of Food and Drug Safety in Korea. However, EK capsule made by Kyunghee University Korean Medicine Hospital, is not separately registered. One EK capsule contains 400 mg of finely grounded EK.

Comment 4. The same applies to the drug Gami-Samhwang-san. Please provide the plant species name in Latin for a complete understanding of the composition of the preparation. What is the content of active ingredients in this drug?

A) Line 321 described the composition by plant species Latin name, “Gami-Samhwang-san (Ephedra Herba, Armeniacae Semen, Acori Gramineri Rhizoma, Raphani Semen, Coicis Semen, Phellodendri Cortex, Atractylodes Chinensis Rhizome, Rhei Radix et Rhizoma)”.

  In our study, two groups were ‘only Gami-samhwang-san’ group’ and ‘EK therapy plus Gami-smhwang-san group’. Gami-Samhwang-san was prescribed to treat obesity in both groups. Because the decoction has 9 herbs, there are too many to figure out active ingredients. However, in aspect of anti-obesity, ephedrine in Ephedra Herba, could be most popular active ingredient.

Comment 5. It is not clear from the manuscript why the authors chose Gami-Samhwang-san as a comparison drug. Perhaps it would be worth adding in the Introduction references for the use of this drug or its components in the treatment of obesity.

A) As we answered above, Gami-Samhwang-san which was prescribed to treat obesity in both groups, was not comparison drug versus EK therapy. Our two study groups were ‘only Gami-samhwang-san’ group’ and ‘EK therapy plus Gami-smhwang-san group’.

Comment 6. What dosage of the drug Gami-Samhwang-san was used?

A) We added dosage of Gami-Samhwang-sanin section 1.2.

Round 2

Reviewer 1 Report

Comments and Suggestions for Authors

I differ with the author, since as he mentions ultrasensitive CRP detects inflammatory processes at extremely low levels, on the contrary the use of AST, ALT and creatinine, are elevated when there is already liver damage, in addition they forgot to complement with bilirubins, gamma glutamyl transferase and even a liver biopsy (gold standard), in addition to creatinine is elevated when renal damage is already declared, therefore it is not a good renal biomarker. So consider the above.

The answer is not satisfactory, since other laboratory test options to evaluate renal dysfunction should have been investigated beforehand, and that was creatinine.

Author Response

Comment 1: I differ with the author, since as he mentions ultrasensitive CRP detects inflammatory processes at extremely low levels, on the contrary the use of AST, ALT and creatinine, are elevated when there is already liver damage, in addition they forgot to complement with bilirubins, gamma glutamyl transferase and even a liver biopsy (gold standard), in addition to creatinine is elevated when renal damage is already declared, therefore it is not a good renal biomarker. So consider the above.

The answer is not satisfactory, since other laboratory test options to evaluate renal dysfunction should have been investigated beforehand, and that was creatinine.

Answer: While ultrasensitive PCR can detect inflammatory processes that may occur before organ damage, you are correct in pointing out that biomarkers like AST, ALT, and creatinine typically become elevated only when there is some degree of liver or kidney damage.

We also acknowledge that a more comprehensive panel of tests, including additional biomarkers like bilirubin, gamma-glutamyl transferase (GGT), and advanced methods such as liver biopsy, could offer deeper insights. Furthermore, while creatinine is a standard measure of renal function, incorporating more sensitive markers could facilitate earlier detection of renal dysfunction.

However, the decision to use AST, ALT, and creatinine in this study instead of ultrasensitive PCR was based on the practicality of these tests and their relevance to assessing organ function in the context of clinical safety.

We appreciate your suggestions and will consider them to enhance the robustness of our safety assessments in future studies.

Round 3

Reviewer 1 Report

Comments and Suggestions for Authors

Please include the response you gave me to the observation I made in the weaknesses of your article.

Author Response

Comment 1: Please include the response you gave me to the observation I made in the weaknesses of your article.

Answer: As your comment, we added the limitation in the discussion section as below;

Finally, the biomarkers used in this study, such as AST, ALT, and creatinine, are typically only activated when the liver or kidneys have been damaged to some degree, therefore, ultrasensitive PCR methods are needed to detect inflammatory processes that may occur before organ damage. Future studies may also incorporate more sensitive markers than bilirubin, gamma-glutamyl transferase (GGT), liver biopsy, and creatinine to help detect liver and kidney dysfunction at an earlier stage.